# Temporal Dynamics of Pulmonary Fibrosis and Immune Dysregulation in a Collagen V-Driven Systemic Sclerosis Model

**DOI:** 10.3390/ijms27010197

**Published:** 2025-12-24

**Authors:** Vitória Elias Contini, Zelita Aparecida J. Queiroz, Sérgio Catanozi, Antonio dos Santos Filho, Lizandre Keren Ramos da Silveira, Aritania Sousa Santos, Sandra de Morais Fernezlian, Denise Frediani, Thays de Matos Lobo, Jaíne Alves Almeida, Camila Machado Baldavira, Ana Paula Pereira Velosa, Percival Degrava Sampaio-Barros, Vera Luiza Capelozzi, Walcy Rosolia Teodoro

**Affiliations:** 1Divisão de Reumatologia, Faculdade de Medicina FMUSP, Universidade de São Paulo, Sao Paulo 01246-903, SP, Brazil; veliascontini@gmail.com (V.E.C.); zelitaaparecida@gmail.com (Z.A.J.Q.); antoniodossantosfilho1965@gmail.com (A.d.S.F.); lizandre_keren@usp.br (L.K.R.d.S.); thaysmatos@usp.br (T.d.M.L.); jaine-almeida@usp.br (J.A.A.); pdsampaiobarros@uol.com.br (P.D.S.-B.); 2Laboratório de Lipides (LIM-10), Hospital das Clinicas HCFMUSP, Faculdade de Medicina FMUSP, Universidade de São Paulo, Sao Paulo 01246-903, SP, Brazil; 3Laboratório de Carboidratos e Radioimunoensaio (LIM 18), Hospital das Clinicas HCFMUSP, Faculdade de Medicina FMUSP, Universidade de São Paulo, Sao Paulo 01246-903, SP, Brazil; aritania@alumni.usp.br; 4Departamento de Patologia, Faculdade de Medicina FMUSP, Universidade de São Paulo, Sao Paulo 01246-903, SP, Brazil; sandra.f@fm.usp.br (S.d.M.F.); ca.mbaldavira@gmail.com (C.M.B.); vera.capelozzi@fm.usp.br (V.L.C.); 5Departamento de Emergência (LIM 51), Faculdade de Medicina FMUSP, Universidade de São Paulo, Sao Paulo 01246-903, SP, Brazil; denisedna@hotmail.com; 6Divisão de Reumatologia, Hospital das Clinicas HCFMUSP, Faculdade de Medicina FMUSP, Universidade de São Paulo, Sao Paulo 01246-903, SP, Brazil; apvelosa@gmail.com

**Keywords:** lung, SSc experimental model, collagen V, systemic sclerosis

## Abstract

Systemic sclerosis (SSc) is a complex autoimmune disease characterized by progressive fibrosis and immune dysregulation, with lung involvement being a major cause of morbidity and mortality. Type V collagen (COLV), a cryptic self-antigen, has been implicated in the pathogenesis of fibrosis in both SSc and lung allograft dysfunction. To characterize the early histological, molecular, and immunological events associated with lung remodeling following immunization with COLV in a murine model (IMU-COLV), and to establish a temporal framework for fibrosis progression. Using a time-course design, lung tissue from IMU-COLV mice was analyzed at multiple intervals post-immunization. Histopathological assessment, immunohistochemistry, and gene expression analysis were performed to evaluate inflammation, endothelial activation, extracellular matrix remodeling, and collagen composition. We observed a progressive and spatially organized pattern of lung remodeling, beginning with peribronchovascular immune infiltration and culminating in airway-centered fibrosis. These changes were accompanied by dynamic endothelial activation, increased expression of profibrotic markers, and alterations in collagen architecture particularly involving COLV. The remodeling pattern closely mirrors histological features observed in early SSc-associated interstitial lung disease and other fibrotic conditions, such as idiopathic pulmonary fibrosis and chronic lung allograft dysfunsion. The IMU-COLV model recapitulates key early features of SSc-related lung fibrosis, highlighting COLV’s potential role as a driver of immune-mediated tissue remodeling. These findings provide a valuable platform for investigating the mechanisms underlying fibrogenesis and for testing targeted interventions in the early phases of pulmonary fibrosis.

## 1. Introduction

Systemic sclerosis (SSc) is a complex autoimmune disease characterized by a combination of vascular dysfunction, fibrosis, and immune system dysregulation. This multifactorial nature leads to diverse symptoms and complications, making SSc particularly challenging to manage and treat [1,2]. Among the most severe manifestations are pulmonary complications, especially interstitial lung disease (ILD), which is marked by progressive scarring of lung tissue. This process impairs pulmonary function and can lead to respiratory failure. In parallel, vascular abnormalities such as pulmonary arterial hypertension further worsen prognosis. These pulmonary manifestations are the leading cause of morbidity and mortality in patients with SSc, highlighting the need for early detection and targeted management strategies [1,2,3].

Despite extensive research, the precise initiating events of SSc remain unclear, and the underlying mechanisms driving fibrosis continue to be poorly defined. To address this, animal models have been instrumental in unraveling aspects of SSc pathophysiology [4,5]. We have previously characterized the IMU-COLV mouse model—induced by immunization with collagen V (COLV)—which replicates key features of human SSc, including cutaneous, vascular, and pulmonary remodeling [6]. This model has emerged as a valuable tool for studying disease progression and evaluating potential therapeutic targets.

COLV is a minor fibrillar collagen localized in the interstitium and capillary basement membranes of pulmonary tissue [7]. Structurally, it consists of the [α1(V)2, α2(V)] heterotrimer, which participates in the formation of heterotypic collagen fibrils, modulating fibril diameter and extracellular matrix (ECM) architecture [8,9]. COLV also influences ECM stiffness and may affect cell behavior during tissue remodeling [10]. Furthermore, due to its normally sequestered location within collagen I/III fibrils, COLV can become immunogenic upon exposure, acting as a potential autoantigen in chronic inflammatory or autoimmune conditions [11,12,13,14,15,16,17,18,19]. Anti-COLV immune responses—particularly against α1(V) chain peptides—have been increasingly reported in early-stage SSc [19,20,21,22,23], and elevated COLV expression has been detected in skin and lung tissues of affected individuals [24,25,26,27,28].

Although our previous work demonstrated that the IMU-COLV model recapitulates pulmonary abnormalities such as increased elastance, airway resistance, arterial remodeling, and interstitial fibrosis—accompanied by increased COLV expression—the temporal dynamics and spatial distribution of these early fibrotic changes have not yet been systematically characterized [6].

In this study, we sought to characterize the early histological, molecular, and immunological changes that occur in the lungs during the onset of fibrosis in the IMU-COLV model. Rather than establishing direct causality, we aimed to describe the temporal and spatial features of lung remodeling associated with COLV immunization. By identifying these early alterations, we intend to generate new hypotheses and lay the groundwork for future studies exploring the mechanisms underlying pulmonary fibrosis in SSc.

## 2. Results

### 2.1. Histological Evaluation of Lung Architecture Using Panoramic Sections

Representative panoramic views of lung tissue stained with Hematoxylin & Eosin (H&E) and Masson’s Trichrome are shown in Appendix A. In control animals, H&E staining revealed preserved alveolar architecture, characterized by thin alveolar septa and absence of inflammatory infiltration. In the IMU-COLV15 group, early alterations were observed, including mild thickening of alveolar walls and discrete peribronchial inflammatory infiltrates, while overall lung architecture remained largely intact.

By 30 days post-immunization (IMU-COLV30), histological changes became more pronounced, with evident peribronchovascular inflammation, thickened alveolar septa, and initial signs of interstitial infiltration extending into the parenchyma. In the IMU-COLV45 group, advanced structural disruption was noted, including dense peribronchovascular and parenchymal infiltrates, alveolar distortion, and extensive fibrotic remodeling, indicating progression to established pulmonary fibrosis.

Masson’s Trichrome staining further highlighted the evolution of fibrotic changes across timepoints. While control lungs displayed minimal collagen deposition, IMU-COLV15 sections showed subtle perivascular and peribronchial collagen accumulation. In IMU-COLV30, collagen fibers became more prominent and extended into adjacent interstitial regions. At 45 days (IMU-COLV45), marked and disorganized collagen deposition surrounded bronchi, vessels, and alveolar structures, consistent with advanced fibrosis. Airway structures were more distinctly visualized in trichrome-stained sections, facilitating a more accurate evaluation of bronchial wall thickening and peribronchial fibrotic remodeling.

Collectively, these panoramic histological analyses provide a clear visualization of the spatiotemporal progression of lung injury in the IMU-COLV model—from early inflammatory remodeling to extensive fibrosis—reinforcing the concept of airway-centered interstitial pathology characteristic of immune-mediated fibrotic lung disease.

### 2.2. Peribronchovascular Inflammation and Early Fibrosis Following Collagen V Immunization

At 15 days post-immunization with COLV (IMU-COLV15), lung tissue exhibited a dense inflammatory infiltrate predominantly in the peribronchial and perivascular regions, as revealed by H&E staining (Figure 1A). This infiltrate extended into the alveolar septa and surrounding small vessels and bronchioles, imparting a strong basophilic hue due to hematoxylin uptake, in contrast to the minimal cellularity observed in the CT group (Figure 1D). Despite the robust immune response, the bronchovascular architecture remained relatively intact, suggesting early inflammation without overt fibrotic disruption.

These histological findings were supported by biochemical quantification of 4-hydroxyproline, which demonstrated a significant increase in total collagen content in the IMU-COLV45 group (5.203 ± 1.350) compared to IMU-COLV15 (3.773 ± 1.006, *p* = 0.0091) and IMU-COLV30 (3.213 ± 0.4715, *p* = 0.0001) (Figure 1M). This progressive increase in collagen supports the transition from inflammation to fibrosis, particularly around the airways and vasculature.

Quantification of immune cell populations confirmed increased TCD3+ lymphocytes in the IMU-COLV15 group compared with both IMU-COLV30 (12.65 ± 4.78 vs. 8.47 ± 2.14, *p* = 0.0058) and IMU-COLV45 groups (12.65 ± 4.78 vs. 8.39 ± 1.68, *p* = 0.0047) (Figure 2E). TCD4+ lymphocytes were also significantly higher in the IMU-COLV15 group than in the IMU-COLV30 (33.99 ± 2.01 vs. 23.38 ± 2.03, *p* = 0.0001) and IMU-COLV45 groups (33.99 ± 2.01 vs. 23.77 ± 0.74, *p* = 0.0001) (Figure 2J). Similarly, TCD8+ lymphocytes were elevated in the IMU-COLV15 group compared with both the IMU-COLV30 (39.40 ± 2.43 vs. 22.87 ± 2.09, *p* = 0.0001) and IMU-COLV45 groups (39.40 ± 2.43 vs. 23.07 ± 2.47, *p* = 0.0001) (Figure 2O). Finally, the number of CD20+ B lymphocytes was significantly higher in the IMU-COLV15 group than in the IMU-COLV30 (34.14 ± 3.81 vs. 27.50 ± 3.22, *p* = 0.0011) and IMU-COLV45 groups (34.14 ± 3.81 vs. 28.24 ± 1.64, *p* = 0.0044) (Figure 2T).

### 2.3. Endothelial Activation and Vascular Remodeling in the IMU-COLV Model

To investigate the vascular component of pulmonary remodeling in the IMU-COLV model, we evaluated the temporal expression of endothelial and vascular markers—Factor VIII, VEGF, α-smooth muscle actin (*α-SMA*), and cleaved caspase-3—across the disease course. Endothelial activation was predominantly localized to the perivascular region; however, VEGF-positive bronchiolar epithelial cells and peribronchial inflammatory cells were also observed, reflecting the close anatomical and functional interplay between airway and vascular compartments. Quantification was restricted to predefined anatomical ROIs relevant to each marker (perivascular microvessels for Factor VIII and caspase-3, peribronchovascular regions for VEGF, and area fraction for *α-SMA*), thereby avoiding distortions associated with whole-section counting.

At 15 days post-immunization (IMU-COLV15), immunostaining for Factor VIII revealed a pronounced increase in endothelial cell activation, particularly within the perivascular microvasculature, indicating early endothelial engagement and possible neovascular recruitment (Figure 3A). This expression markedly declined at 30 and 45 days (Figure 3B,C), reaching levels comparable to non-immunized control lungs (Figure 3D). Quantitative analysis confirmed significantly elevated Factor VIII expression in the IMU-COLV15 group compared to both IMU-COLV30 (13.87 ± 0.426 vs. 5.16 ± 0.0983, *p* = 0.0001) and IMU-COLV45 (13.87 ± 0.426 vs. 4.99 ± 0.90, *p* = 0.0001) (Figure 3E).

A similar temporal profile was observed for VEGF expression, which was highest in the IMU-COLV15 group. VEGF+ endothelial cells were predominantly detected in inflamed and peribronchovascular regions, suggesting active angiogenic signaling during early tissue injury. Expression decreased progressively over time, consistent with the transition from a pro-angiogenic to a fibrotic environment. Quantification demonstrated significantly reduced VEGF levels at later timepoints (IMU-COLV15 vs. IMU-COLV30: 3.81 ± 0.78 vs. 2.64 ± 0.57, *p* = 0.0008; IMU-COLV15 vs. IMU-COLV45: 3.81 ± 0.78 vs. 2.45 ± 0.73, *p* = 0.0001) (Figure 3J).

In contrast, *α-SMA* staining, initially restricted to the smooth muscle layers of vascular and bronchial walls, showed a progressive increase in both intensity and thickness, particularly in the IMU-COLV45 group (Figure 3F–H). This pattern reflects perivascular and bronchial smooth muscle hyperplasia and structural reinforcement, characteristic of chronic vascular remodeling. Quantitative analysis confirmed a significant increase in *α-SMA* expression in IMU-COLV45 compared to IMU-COLV15 (29.45 ± 5.39 vs. 23.43 ± 2.52, *p* = 0.0051) and IMU-COLV30 (29.45 ± 5.39 vs. 24.54 ± 2.31, *p* = 0.0421) (Figure 3O).

Finally, cleaved caspase-3 immunostaining revealed a late-onset increase in endothelial apoptosis, most prominent in the microvasculature of IMU-COLV45 lungs (Figure 3R,S). This finding is indicative of persistent vascular injury and endothelial cell turnover during advanced fibrosis. Quantitative data showed significantly higher caspase-3+ endothelial cell density in IMU-COLV45 compared to IMU-COLV30 (13.60 ± 2.76 vs. 9.89 ± 1.79, *p* = 0.0161) (Figure 3T).

Collectively, these findings illustrate a dynamic, temporally coordinated process of vascular involvement in the IMU-COLV model—beginning with early endothelial activation (Factor VIII and VEGF), progressing to vascular remodeling (*α-SMA* thickening), and culminating in endothelial cell loss (caspase-3+ apoptosis). These events are spatially centered on the bronchovascular compartments and align with histological regions of inflammation and fibrosis, reinforcing the airway-centered pathogenesis observed in this model of immune-mediated pulmonary fibrosis.

### 2.4. Collagen Composition and Matrix Remodeling in Peribronchovascular Fibrosis in the IMU-COLV Model

To characterize extracellular matrix remodeling in the IMU-COLV model, we assessed the spatiotemporal distribution of collagen types I, III, and V, with emphasis on the bronchovascular compartments. To ensure specificity, all immunolabeling procedures were accompanied by isotype IgG controls and no-primary-antibody controls, which confirmed that the fluorescence patterns observed for collagens I, III, and V reflected true antigen-specific staining rather than nonspecific background signal.

At 15 days post-immunization (IMU-COLV15), type I collagen (Col I) was sparsely distributed and mainly localized to the perivascular and peribronchial regions, indicating initial remodeling with preserved parenchymal architecture (Figure 4A). By day 30 (IMU-COLV30), Col I deposition increased significantly in these regions, resulting in visible thickening of both vascular and bronchial walls when compared to control animals (Figure 4B vs. Figure 4D). At 45 days (IMU-COLV45), Col I accumulation became extensive and disorganized, extending into the alveolar septa and surrounding interstitium (Figure 4C), reflecting advanced fibrotic remodeling and architectural distortion. Quantitative analysis confirmed a progressive increase in Col I content, with significantly higher expression in IMU-COLV45 compared to IMU-COLV15 (39.76 ± 7.18 vs. 20.29 ± 4.66, *p* = 0.0240), consistent with the transition from early matrix deposition to mature fibrosis.

Type III collagen (Col III) followed a comparable temporal pattern. In the IMU-COLV15 group, Col III distribution was limited to the peribronchovascular areas and resembled that of control lungs (Figure 4F,I). At 30 days, however, Col III expression expanded from the bronchovascular axis into neighboring interstitial zones (Figure 4G), indicative of active fibrotic remodeling. By 45 days, Col III staining was diffusely distributed throughout the lung interstitium, co-localizing with Col I in fibrotic foci (Figure 4H). This widespread deposition parallels the histological findings of interstitial thickening and loss of alveolar integrity. Quantitative data confirmed a significant increase in Col III expression in IMU-COLV30 (31.04 ± 7.20, *p* = 0.0059) and IMU-COLV45 (30.53 ± 4.01, *p* = 0.0107) compared to IMU-COLV15 (21.67 ± 4.41) (Figure 4J).

Interestingly, type V collagen (COLV), the antigen used for immunization, exhibited a distinct temporal profile. At 15 days post-immunization, COLV was abundantly expressed as fine fibrillar structures along peribronchovascular areas and within the pulmonary microvasculature (Figure 4K), suggesting its early involvement in immune targeting and structural disruption. Although COLV fibers were still detectable at 30 and 45 days (Figure 4L–N), quantitative analysis revealed a significant decline in its expression over time. Compared to IMU-COLV15, COLV levels were reduced in IMU-COLV30 (9.24 ± 1.28 vs. 11.26 ± 1.24, *p* = 0.0181) and further decreased in IMU-COLV45 (8.65 ± 1.34, *p* = 0.0031), indicating possible degradation or loss of this native matrix component as fibrosis progressed. These findings suggest that the autoimmune response against COLV may not only drive inflammation but also contribute to structural matrix instability, thereby facilitating fibrotic remodeling.

Collectively, these results reveal a coordinated progression of peribronchovascular fibrosis in the IMU-COLV model, marked by increasing Col I and III deposition and disorganization, alongside a loss of structural COLV. This pattern supports the concept of airway-centered interstitial remodeling and mirrors pathological mechanisms implicated in immune-mediated fibrotic lung diseases.

### 2.5. Gene Expression Profile of Collagen and Fibrosis-Related Markers in Lung Tissues of the IMU-COLV Model

To explore the molecular mechanisms underlying fibrotic remodeling in the IMU-COLV model, we performed quantitative RT-PCR analysis of lung tissue mRNA, focusing on key pro-fibrotic mediators and collagen-encoding genes. These included the α1 chains of collagen types I and III (*Col1a1* and *Col3a1*), the α1 and α2 chains of collagen V (*Col5a1* and *Col5a2*), as well as classical markers of fibroblast activation and epithelial–mesenchymal transition (EMT).

*Tgfb1* (Transforming growth factor beta 1) gene, that regulates TGF-β, a master regulator of fibrogenesis and central to systemic sclerosis (SSc) pathogenesis, was significantly upregulated at day 45 post-immunization (IMU-COLV45) compared to both IMU-COLV15 and IMU-COLV30 (2.46 ± 0.81 vs. 0.93 ± 0.64, *p* < 0.0001; vs. 1.16 ± 0.84, *p* = 0.0003) (Figure 5A). This increase coincided with marked elevations in the *Acta2* (α-Smooth muscle actin) gene expression (2.79 ± 0.78 vs. 1.03 ± 0.45 and 1.43 ± 0.36; *p* < 0.0001 for both) (Figure 5B), and *Vim* (vimentin), another mesenchymal marker, which also showed significant upregulation at IMU-COLV45 (2.24 ± 0.38 vs. 1.09 ± 0.48 and 1.41 ± 0.37; *p* < 0.0001 for both) (Figure 5C). These findings point to sustained fibroblast activation and EMT as the disease progresses.

In contrast, expression of collagen V genes demonstrated a declining pattern over time, mirroring the immunohistochemical findings. *Col5a1* was significantly reduced in IMU-COLV30 and IMU-COLV45 compared to IMU-COLV15 (1.29 ± 0.48 and 1.86 ± 0.30 vs. 2.78 ± 0.76; *p* < 0.0001 and *p* = 0.0093, respectively) (Figure 6E). Similarly, *Col5a2* expression decreased from IMU-COLV15 to both IMU-COLV30 (1.02 ± 0.73) and IMU-COLV45 (1.81 ± 0.52), when compared to early overexpression at day 15 (2.94 ± 0.88; *p* < 0.0001 and *p* = 0.0065, respectively) (Figure 5G).

Collectively, these molecular data demonstrate a temporal shift in gene expression during the course of lung fibrosis in the IMU-COLV model. The upregulation of pro-fibrotic mediators (*TGF-β*, *α-SMA*, *Vimentin*) and fibrillar collagens (*Col1a1*, *Col3a1*), alongside the downregulation of native matrix components (*Col5a1*, *Col5a2*), reflects an imbalance between matrix synthesis and degradation. This molecular profile underscores a pathogenic transition from immune-mediated injury to established fibrotic remodeling, resembling key features of fibrosing interstitial lung diseases such as systemic sclerosis-associated interstitial lung disease (SSc-ILD).

### 2.6. Detection of Antinuclear Antibodies (ANA) During Fibrosis Progression in the IMU-COLV Model

To investigate the presence of autoantibodies associated with autoimmune processes, we performed indirect immunofluorescence (IIF) for antinuclear antibodies (ANA) using HEp-2 cells and sera collected from control (CT) and IMU-COLV animals at different timepoints. No significant differences in ANA reactivity were observed between IMU-COLV15 and CT groups, indicating minimal or absent autoantibody production at early stages of the disease. However, a marked increase in ANA positivity was detected at later timepoints, with significantly higher frequencies in the IMU-COLV30 (*p* = 0.0017) and IMU-COLV45 groups (*p* = 0.0006), reflecting progressive autoimmune activation in parallel with fibrotic remodeling (Figure 6A). Representative images illustrate a lack of nuclear fluorescence in ANA-negative sera from control animals, in contrast to the pronounced nuclear staining observed in ANA-positive sera from IMU-COLV animals (Figure 6B).

## 3. Discussion

In this study, we employed a time-course approach to investigate the histopathological and molecular events associated with lung inflammation and fibrosis induced by immunization with collagen type V (COLV). Our findings reveal clear temporal progression from early peribronchovascular inflammation to established fibrosis, accompanied by dynamic changes in endothelial activation, ECM remodeling, and collagen composition. One of the major strengths of our study is the use of the IMU-COLV model for investigating pulmonary fibrosis evolution in SSc, particularly in its early stages. Although several experimental SSc models have been developed to examine immune dysregulation and fibrosis, a comprehensive understanding of fibrosis progression over time in these models remains elusive [29]. In this regard, the IMU-COLV model offers a unique advantage, as it demonstrates early immune infiltration and pulmonary fibrosis within 15 days post-induction, an observation that distinguishes it from other commonly used models.

The modifications in the constitution of pulmonary fibrosis in the IMU-COLV model were previously observed by Teodoro et al. [6]; these changes closely resemble the morphological pattern of interstitial fibrosis observed in patients with SSc. In both animals and humans with SSc, in addition to the increased collagen synthesis, structurally anomalous COLV was identified, characterized by shorter, thicker fibers with heterogeneous distribution [24]. Studies have demonstrated that patients with SSc exhibit increased COLV, *Col5a2* gene, and α2(V) chain expressions in both skin and lung tissues during the early stages of the disease [26,27,28]. Moreover, anomalous COLV deposition in the skin of patients with SSc is associated with skin thickening and disease activity [28]. These findings, coupled with the observation that nasal COLV tolerance in a rabbit scleroderma model reduces both the inflammatory and fibrotic processes in the lung and skin, along with a decrease in TGF-β expression, further support the hypothesis that COLV plays a pivotal role in SSc pathogenesis. COLV acts as a neoantigen that triggers an autoimmune response, contributing to disease progression [30,31].

Our study demonstrates that immunization with collagen type V (COLV) leads to a progressive inflammatory response that originates peribronchovascularly and ultimately culminates in airway-centered fibrosis. This process closely resembles fibrotic mechanisms observed in diseases such as chronic lung allograft dysfunction [32] and idiopathic pulmonary fibrosis [33]. The peribronchovascular localization and subsequent centripetal progression of inflammation and fibrosis are consistent with previous descriptions of COLV as a cryptic self-antigen primarily located in the peribronchovascular matrix and subendothelial regions. Exposure of COLV through epithelial and endothelial injury has been shown to elicit an autoimmune response, a phenomenon that may underlie graft dysfunction following lung transplantation and progressive airway remodeling in fibrotic lung diseases [34].

The early phase (15 days post-immunization) was characterized by a robust inflammatory infiltrate predominantly localized to peribronchial and perivascular regions, with preservation of alveolar structures. This inflammatory response likely reflects an antigen-specific reaction to COLV, a minor but immunogenic component of the ECM known to elicit autoreactive T-cell responses and contribute to immune-mediated tissue injury [35,36]. As shown in the immunophenotypic panel, this phase was marked by prominent lymphocytic infiltration; however, we also observed mononuclear phagocyte-rich aggregates in H&E sections, suggesting participation of the macrophage compartment. Although macrophage-specific markers (e.g., CD68, F4/80) were not included in the current immunohistochemical panel, their involvement is consistent with the well-established role of macrophages in initiating and amplifying pulmonary fibrotic responses. We acknowledge this as a limitation of the present work and highlight that future studies incorporating macrophage subset characterization, including M1/M2 polarization—will be essential to delineate their contribution to COLV-induced fibrosis. Notably, this early phase showed only mild ECM deposition, with picrosirius red staining revealing a predominance of immature, thin collagen fibers, and Masson’s trichrome confirming minimal fibrosis. These findings align with studies demonstrating that peribronchial inflammation precedes fibrotic remodeling in several models of pulmonary fibrosis [37,38].

Human placental collagen type V (COLV) was selected because it is the most widely used and well-characterized source of purified COLV for experimental models of fibrosis and autoimmunity. Importantly, human COLV shares high structural and immunological homology with murine COLV, which enables cross-reactive immune responses relevant to disease mechanisms. Although human COLV is technically a heterologous antigen, previous studies have consistently demonstrated that the fibrosis and inflammatory profiles resulting from COLV immunization are antigen-specific and not due to nonspecific reactions to a foreign protein. This includes evidence of COLV-specific lymphocytic responses, targeted peribronchovascular inflammation, and collagen-focused tissue remodeling—patterns not reproduced by control immunizations using other heterologous collagens or irrelevant proteins. In our study, the use of human COLV follows these established models, ensuring comparability with prior literature.

By 30 days, inflammation had partially subsided, giving way to fibroproliferative activity, marked by increased deposition of collagen types I and III and progressive thickening of the peribronchovascular matrix. The shift toward orange-red birefringent fibers under polarized light microscopy and the enhanced immunoreactivity for type I and III collagens confirm the transition to an intermediate fibrotic phase with early collagen maturation. Similar remodeling patterns have been described in bleomycin-induced lung fibrosis models and in human idiopathic pulmonary fibrosis (IPF), in which type I collagen dominates the ECM during disease progression [35,36].

At 45 days post-immunization, we observed a fully developed fibrotic phenotype, with dense, disorganized collagen deposition, significant architectural distortion, and near-complete replacement of peribronchovascular regions by fibrotic tissue. This advanced fibrotic stage was associated with increased 4-hydroxyproline content, further supporting the histological evidence of collagen accumulation. *Col1a1*, *Col3a1*, *Vim* (Vimentin), and *Tgfb1* gene upregulation further confirmed the presence of fibrosis, indicating that the fibrotic process is not only sustained but also progressing. TGF-β1 plays a central role in the fibrotic process by promoting fibroblast differentiation into myofibroblasts and inducing the synthesis of collagen and other ECM proteins [37]. The progressive increase in *α-SMA* expression in the bronchial and vascular smooth muscle layers, rather than in interstitial myofibroblasts, suggests that airway remodeling in the IMU-COLV model may be partially driven by smooth muscle hypertrophy and vascular stiffening. These features are consistent with chronic fibrotic processes involving both airway and vascular compartments [38,39]. These results corroborate previous reports highlighting COLV as a driver of fibrosis in both lung and vascular tissue, likely through immune-mediated mechanisms and fibroblast activation [40,41].

Beyond classical *α-SMA*-positive myofibroblasts, a range of activated stromal cells (activated fibroblasts, pericytes and other mesenchymal subpopulations) have been implicated in initiating and perpetuating fibrotic programs in multiple organs, including the lung. In our IMU-COLV model, the progressive increase in *α-SMA* immunoreactivity and coordinated rise in collagen I/III/V deposition are consistent with activation of stromal effector populations; however, *α-SMA* alone cannot discriminate between bona fide myofibroblasts and other contractile or activated stromal phenotypes. We therefore acknowledge that the present study does not provide direct phenotypic identification of distinct stromal subsets. Future work should employ complementary approaches—for example, immunostaining for PDGFRα, fibroblast activation protein (FAP), periostin, prolyl-4-hydroxylase or NG2 (for pericytes), combined with multiplex imaging or single-cell RNA sequencing—to define whether and when specific stromal cell types are mobilized in this model. Such experiments would clarify whether activated stromal cells contribute to early fibrogenesis in the IMU-COLV model and would help map stromal–immune interactions that drive progression to established fibrosis. Future studies will include multiplex immunofluorescence (e.g., PDGFRα, FAP, periostin, NG2) and single-cell transcriptomics to resolve stromal heterogeneity and define the contribution of distinct activated mesenchymal populations to COLV-induced fibrosis.

Interestingly, in our COLV-induced model, the earliest pathological alterations (15 days post-immunization) were indeed predominantly peribronchial and perivascular, with the alveolar interstitium largely preserved. This pattern reflects the known distribution of collagen V, which is enriched in peribronchovascular connective tissue and therefore becomes an early target of autoimmune recognition. As the disease progresses (30–45 days), we observed progressive extension of fibrosis into the alveolar interstitium, confirmed by Masson’s trichrome and Picrosirius red staining, indicating that interstitial thickening develops secondarily, following initial peribronchovascular injury.

One of the most intriguing findings was the temporal pattern of COLV itself. While COLV fibers were detected structurally throughout the time course, their expression was significantly reduced at later stages. This suggests a paradoxical loss or degradation of endogenous COLV in fibrotic lungs, possibly reflecting remodeling of basement membranes and changes in ECM composition. Similar alterations in COLV distribution have been reported in fibrotic skin and lung tissues, where disrupted collagen organization may contribute to the persistence of inflammation and impaired tissue regeneration [42].

The peribronchovascular inflammatory and fibrotic response observed in the IMU-COLV model reflects a biologically plausible and clinically relevant pattern of lung involvement that parallels systemic sclerosis-associated interstitial lung disease (SSc-ILD). Collagen V is a minor fibrillar collagen, primarily located within the interstitial matrix, where it is co-distributed with type I collagen and plays a role as a regulatory component of heterotypic collagen fibrils. It is distributed in the lung interstitium, where it is predominantly localized within peribronchial connective tissue, vascular adventitia, the alveolar interstitium, and subpleural regions. Under physiologic conditions, COLV remains immunologically sequestered; however, tissue injury, oxidative stress, or remodeling events—common features of autoimmune and fibrotic disorders—can expose normally cryptic epitopes and enhance its immunogenicity. In this context, immunization with COLV likely breaks tolerance to a structurally concealed self-antigen, promoting an autoimmune response directed toward anatomical regions naturally enriched in COLV. This spatial correspondence helps explain the selective peribronchial and perivascular inflammation and fibrosis observed in our model and supports the relevance of COLV-targeted immune mechanisms in the pathogenesis of SSc-ILD [13,18,43].

The peribronchovascular predilection of fibrosis in the IMU-COLV model mimics key histopathological features of SSc-ILD, where early fibrotic changes frequently begin in the peribronchial and perivascular interstitium, later progressing to involve alveolar septa. This distribution reflects both antigen localization and vascular vulnerability, hallmarks of SSc pathogenesis, which involves microvascular injury, endothelial dysfunction, and perivascular fibrosis as initiating events [44,45].

Previous studies demonstrate that Col V immunization induces a Th17/Th1-skewed immune response, which has been implicated in fibroblast activation, endothelial cell dysfunction, and profibrotic cytokine production (e.g., TGF-β, IL-6, IL-17) [46]. These cytokines are particularly active in perivascular niches, where immune–stromal interactions amplify fibrotic cascades. Thus, the spatial orientation of fibrosis in the IMU-COLV model is not random but driven by immunologic targeting of Col V-rich regions.

Interestingly, our findings also reveal a sequential pattern of endothelial activation, angiogenesis, and subsequent injury. Elevated expression of Factor VIII and VEGF at day 15 indicates early endothelial activation and hypoxia-driven angiogenic signaling, a response commonly seen in acute lung injury and early fibrotic lung disease [47,48]. However, VEGF expression diminished over time, coinciding with vascular remodeling and increased cleaved caspase-3 expression, suggestive of endothelial apoptosis and vascular regression. This progression mirrors findings in other fibrotic lung models, where initial endothelial activation is followed by vascular dropout and capillary rarefaction, contributing to tissue hypoxia and fibrogenesis [49,50]. The early endothelial activation (Factor VIII, VEGF) and smooth muscle thickening observed in the model indicate that vascular structures are not merely bystanders but active participants in the fibrotic process. This aligns with the “vasculopathy-first” hypothesis in SSc, where microvascular injury precedes fibrotic remodeling [51,52], particularly in bronchovascular bundles, where vessels, nerves, and lymphatics are tightly interwoven with extracellular matrix [53].

Collectively, these findings demonstrate that immunization with COLV triggers a temporally regulated sequence of lung injury events: starting with peribronchovascular inflammation, progressing through ECM remodeling, and culminating in irreversible fibrotic scarring. The localization of changes predominantly around the bronchovascular compartments highlights the importance of these niches in orchestrating fibrotic progression. The IMU-COLV model thus represents a useful tool to study fibrosis initiation and progression with relevance to systemic sclerosis and other immune-mediated interstitial lung diseases.

An important methodological consideration in our study relates to fluorescence imaging. Murine lung tissue is known to exhibit intrinsic autofluorescence, particularly within collagen-rich or inflamed regions, which can increase background signal and partially obscure specific fluorophore detection. Although background subtraction was applied during image acquisition and exposure settings were standardized across groups, autofluorescence could not be fully eliminated. To ensure specificity, all immunolabeling procedures were accompanied by isotype IgG controls and no-primary-antibody controls, which confirmed that the fluorescence patterns observed for collagens I, III, and V reflected true antigen-specific staining rather than nonspecific background signal. Nevertheless, we acknowledge that tissue autofluorescence remains a limitation inherent to lung fluorescence imaging and may modestly influence qualitative visualization, although it does not compromise the validity of the comparative findings reported here.

Limitations of this study include the use of single-marker immunohistochemistry, the absence of flow cytometry or co-staining, and the lack of functional assays (e.g., anti-COLV neutralization or tolerance induction). These limitations prevent us from drawing firm conclusions about the role of COLV in driving fibrosis. Future studies incorporating interventional experiments will be essential to test the causal relevance of COLV and its potential as a therapeutic target. However, despite these limitations, the IMU-COLV model offers a valuable platform for studying early events in SSc lung disease and generating testable hypotheses about the interplay between ECM remodeling, immune activation, and vascular dysfunction in fibrosis.

## 4. Materials and Methods

### 4.1. Experimental Model

Female C57BL/6 mice (*n* = 54), aged 6–7 weeks, were obtained from the animal facility of our institution. Female C57BL/6 mice were used in all experiments due to the known sex-related differences in the incidence and pathophysiology of systemic sclerosis (SSc) and SSc-associated interstitial lung disease (SSc-ILD). These conditions occur predominantly in women, and female mice more reliably reproduce the immune and fibrotic responses associated with this sex bias. Therefore, the selection of females provides a biologically relevant model for studying COLV-induced autoimmunity and fibrosis. The mice were housed in specific pathogen-free conditions with free access to food and fresh water in a temperature-controlled room (22–24 °C) and maintained on a 12 h light/dark cycle. All experimental procedures were approved by The Committee on Ethical Use of Laboratory Animals of the Faculty of Medicine at the University of São Paulo (CEUA-protocol number: 1543/2020; 1416/2019. The experimental study was reported in accordance with ARRIVE guidelines.

To induce experimental SSc, mice (*n* = 30) were subcutaneously immunized with 150 µg of human placental COLV (Col V; Sigma-Aldrich, St. Louis, MO, USA) dissolved in 10 mM acetic acid and emulsified with an equal volume of complete Freund’s adjuvant (Sigma-Aldrich). At day 20, post-immunization, two booster doses of 150 µg/200 µL COLV mixed with incomplete Freund’s adjuvant were administered via intramuscular injection at 15-day intervals (IMU-COLV) (Figure 7). One hour before immunization, tramadol hydrochloride (40 mg/kg body weight) was administered subcutaneously as preventive analgesia.

The control group (CT; *n* = 24) was inoculated with 10 mM acetic acid mixed with complete or incomplete Freund’s adjuvant, following the same immunization protocol. To investigate early pulmonary changes associated with experimental SSc, animals were euthanized at 15 (*n* = 10), 30 (*n* = 10), and 45 (*n* = 10) days after the initial COLV immunization. Before euthanasia, the mice were anesthetized with an intraperitoneal injection of ketamine hydrochloride (100 mg/kg body weight) and xylazine (10 mg/kg body weight), followed by euthanasia through nuchal dislocation.

### 4.2. Preparation

Lung tissue samples were collected and fixed in 10% buffered formalin for morphometric and immunohistochemical analysis. Additional lung fragments were stored at −80 °C for subsequent molecular analysis.

### 4.3. Lung Morphological Analysis

To ensure anatomical consistency and avoid bias due to regional heterogeneity of pulmonary fibrosis, lung sampling was fully standardized across all animals. For all molecular analyses (qPCR and biochemical assays), tissue was consistently collected from the right lung, specifically the middle and inferior lobes, which were dissected in bloc following the same anatomical landmarks. These lobes were chosen because they provide sufficient material for parallel assays and exhibit reproducible inflammatory–fibrotic changes in our model. The same procedure was applied uniformly to all mice, minimizing variability related to lobe-specific differences in fibrosis distribution.

For histological examination, lung samples were fixed in 10% buffered formalin for 24 h and subsequently embedded in paraffin. Sections of 3–4 µM thickness were cut and stained with hematoxylin and eosin (H&E) for general histological assessment, and picrosirius red for collagen fiber detection.

### 4.4. Immunofluorescence for Collagen

For collagen immunostaining (Col I, III, and V), 4 µm lung tissue sections were adhered to glass slides pre-coated with 3-aminopropyltriethoxy silane (Sigma-Aldrich). Antigen retrieval was performed by incubating the slides with bovine pepsin (8 mg/mL in 0.5 N acetic acid) (Sigma-Aldrich) for 30 min at 37 °C. After several washes with phosphate-buffered saline (PBS), the slides were blocked with 5% bovine serum albumin (BSA) diluted in PBS (pH 7.0).

Subsequently, the slides were incubated overnight at 4 °C with rabbit polyclonal antibodies: anti-Col I (1:1200; Rockland Immunochemicals, Inc.; Pottstown, PA, USA), anti-Col III (1:1400; Rockland Immunochemicals), and anti-Col V produced in the laboratory (1:1000) [24]. After washing with PBS containing 0.05% Tween 20, the slides were incubated for 1 h at 25 °C with goat anti-rabbit IgG Alexa Fluor 488 (Invitrogen, Life Technologies, Carlsbad, CA, USA) at a 1:200 dilution in PBS containing 0.006% Evans blue.

The slides were mounted with buffered glycerin and analyzed under a fluorescence microscope (Olympus BX-51, Olympus Co., Tokyo, Japan).

To ensure specificity, all immunolabeling procedures were accompanied by negative controls, which confirmed that the fluorescence patterns observed for collagens I, III, and V reflected true antigen-specific staining rather than nonspecific background signal.

### 4.5. Immunohistochemistry

For immunohistochemical analysis, 3–4 µm lung tissue sections were deparaffinized and incubated in a 0.3% hydrogen peroxide solution for 5 min to inhibit endogenous peroxidase activity. Afterward, sections underwent a cycle of four 5 min washes.

The following primary antibodies were used for the immunostaining reactions: smooth muscle α-actin (*α-SMA*, mouse monoclonal, 1:50; Santa Cruz Biotechnology, Inc.; Dallas, TX, USA), vascular endothelial growth factor (VEGF, 1:400; Santa Cruz Biotechnology Inc.), Factor VIII (1:1200; Santa Cruz Biotechnology Inc.), Caspase-3 (1:4000; Novus Biologicals, Centennial, CO, USA), CD3 (1:1000; Santa Cruz Biotechnology Inc.), CD4 (1:1000; Santa Cruz Biotechnology Inc.), CD8 (1:1200; Santa Cruz Biotechnology Inc.) and CD20 (1:100; Santa Cruz Biotechnology Inc.).

Antigen retrieval was performed using either bovine pepsin (4 mg/mL in 0.01 N glycine buffer, Sigma-Aldrich) for 30 min at 37 °C, or citrate buffer (pH 6.0) at high temperature (125 °C for 1 min) in a pressure cooker Pascal (Dako, Carpinteria, CA, USA). Subsequently, the sections were incubated with the primary antibodies overnight at 4 °C.

Immunoreactions were detected using a biotin–streptavidin–peroxidase kit (Vector Laboratories; Newark, CA, USA) per manufacturer instructions. The chromogen used was 3,3′-diaminobenzidine (Sigma-Aldrich), and the sections were counterstained with Harris hematoxylin (Merck KGaA, Darmstadt, Germany).

Notably, our staining protocol included negative controls processed in parallel, which confirmed the specificity of the immunolabeling for CD3, CD4, CD8, CD20, Factor VIII, Caspase-3 and VEGF. 

### 4.6. Morphometric Analysis

Collagen fibers of types I, III, and V were quantified in the lung parenchyma using Image-Pro Plus 6.0 software (Media Cybernetics, Rockville, MD, USA). Two blinded observers analyzed 10 randomly selected microscopic fields at 400× magnification. Collagen fiber thresholds were uniformly set on all slides after contrast enhancement, ensuring that fibers were reliably visualized as green bands. Collagen fiber density was calculated as the ratio of the measured collagen-positive area to the total tissue area analyzed, expressed as a percentage.

For immunostained cell analysis, 10 random lung tissue fields were evaluated at 1000× magnification using Image-Pro Plus 6.0 software. The expression of *α-SMA*, VEGF, Factor VIII, caspase-3, CD3, CD4, CD8, and CD20 was quantified by counting the number of positive cells and expressing this value as the proportion of total cells in each field.

To minimize bias arising from cell clustering or regional heterogeneity, the quantification of positive cells was performed using a standardized region-of-interest (ROI) strategy with anatomically predefined compartments applied consistently across all samples. Factor VIII-positive cells were counted exclusively within perivascular microvessels. VEGF-positive cells were quantified in perivascular and peribronchial ROIs; epithelial VEGF positivity was recorded qualitatively but excluded from numerical quantification. *α-SMA* immunostaining was quantified as the stained area fraction within the peribronchovascular compartment rather than by cell counting. Cleaved caspase-3-positive cells were quantified only in endothelial cells within microvessels, identified morphologically and by anatomical localization. All analyses were performed in ImageJ (version 1.53, National Institutes of Health, Bethesda, MD, USA) using automated thresholding with subsequent manual verification by two independent blinded observers. For each mouse, three to five non-overlapping ROIs were analyzed at 400× magnification.

### 4.7. Quantitative Real-Time Polymerase Chain Reaction (qRT-PCR)

The total RNA was isolated according to a standard Trizol^®^ (Invitrogen, Life Technologies Co., Carlsbad, CA, USA) RNA isolation protocol. Table 1 shows the sequence of oligonucleotides of the *Col1a1* (Col I α1 chain), *Col3a1* (Col III α1 chain), *Col5a1* (Col V α1 chain), *Col5a2* (Col V α2 chain), *Tgf1b* (TGF-β), *Vim* (Vimentin), and *Acta2* (*α-SMA*) genes. The genes expressions were assessed via qRT-PCR.

For cDNA synthesis, total RNA from each sample was reverse-transcribed using the High-Capacity cDNA Reverse Transcription Kit (Applied Biosystems, Foster City, CA, USA). qRT-PCR reactions were prepared using the Platinum SYBR Green qPCR SuperMix-UDG Kit (Invitrogen, Life Technologies). Amplification and cDNA production were performed using a Step One thermal cycler (Applied Biosystems), with 1000 ng of total RNA per sample.

Relative gene expression was calculated using the 2^−ΔΔCT^ method, with β-2 microglobulin (*B2m*) used as the housekeeping gene for normalization.

### 4.8. Collagen Quantification by Measuring 4-Hydroxyproline

Total lung collagen deposition was quantified by measuring the 4-hydroxyproline content, as previously described with modifications [54]. Briefly, lung samples were freeze-dried (Edwards Vacuum, West Sussex, UK), weighed, and subsequently hydrolyzed in 6 N HCl for 22 h at 100 °C. The hydroxyproline content was determined spectrophotometrically by measuring absorbance at 560 nm. The results were expressed as nanograms of 4-hydroxyproline per milligram of protein [54].

### 4.9. Anti-Nuclear Antibody (ANA) Detection

Sera obtained from IMU-COLV and CT groups of mice from 15, 30, and 45 days were tested for ANAs using slides with HEp-2 cells (Nova-Lite HEp-2 ANA Kit; Inova Diagnostics, San Diego, CA, USA), per manufacturer instructions. Slides were incubated for 30 min at room temperature with sera diluted at 1:80 in PBS with Tween20. After a washing cycle with PBS/Tween20, the slides were incubated for 30 min with conjugated goat anti-mouse IgG antibody ALEXA FLUOR 488 (Invitrogen, Life Technologies) (1:200). Subsequently, slides were counterstained with 0.006% Evans blue, mounted on coverslips with buffered glycerin and examined under a fluorescence microscope (Olympus BX-51).

### 4.10. Statistical Analysis

Data were analyzed for normality of distribution and categorized as parametric or non-parametric accordingly, using GraphPad Prism version 8.0.2 (GraphPad Software, San Diego, CA, USA). Results for each variable are presented as means ± standard deviation. One-way analysis of variance was used to compare the means between groups, with Tukey or Sidak’s post-test applied for normally distributed variables. The Kruskal–Wallis test was used for non-normally distributed variables, followed by the Dunn test.

## Figures and Tables

**Figure 1 ijms-27-00197-f001:**
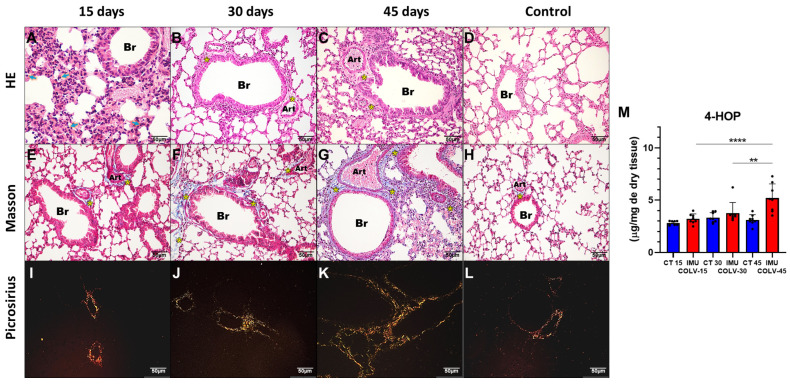
Histological and Collagen Deposition Patterns in Lung Tissue Following COLV Immunization. Representative images of lung sections stained with Hematoxylin & Eosin (H&E), Masson’s Trichrome, and Picrosirius Red at 15, 30, and 45 days after COLV immunization (IMU-COLV), compared with non-immunized controls (CT). H&E staining (**A**–**D**) illustrates inflammatory changes (blue arrows) and airway-associated remodeling. Masson’s Trichrome (**E**–**H**) highlights collagen deposition (yellow asterisks) and structural alterations in peribronchial and perivascular regions. Picrosirius Red staining under polarized light (**I**–**L**) distinguishes collagen fiber maturation, showing thin (green–yellow) and thick (orange–red) birefringent fibers. Quantification of total collagen content by 4-hydroxyproline assay is shown in (**M**). ** *p* < 0.01; **** *p* = 0.0001. Note: Bronchiolar structures naturally vary in size across sections; this variability is acknowledged as a limitation in comparative visualization.

**Figure 2 ijms-27-00197-f002:**
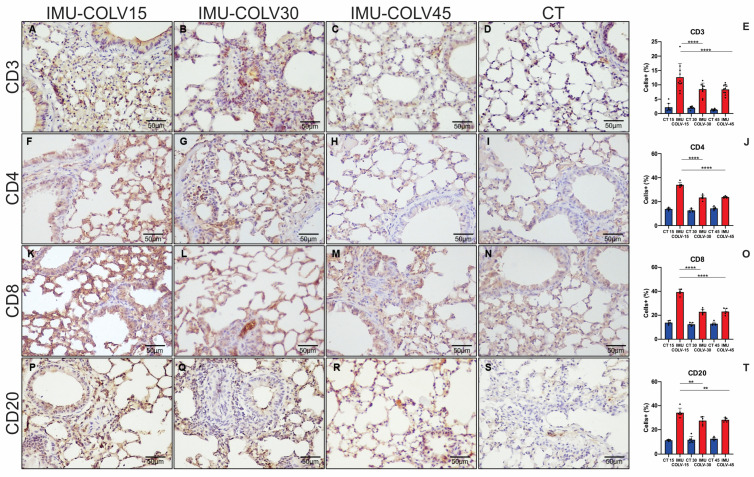
Histomorphometric analysis of lymphocytic inflammatory cell infiltration in lung tissue. Representative photomicrographs illustrate the expression of CD3, CD4, CD8, and CD20 in lung sections at 15 days post-immunization (**A**,**F**,**K**,**P**), 30 days (**B**,**G**,**L**,**Q**), and 45 days (**C**,**H**,**M**,**R**). A progressive reduction in the expression of all evaluated lymphocytic markers was observed over time, as confirmed by the corresponding quantitative analyses (**E**,**J**,**O**,**T**). The control group demonstrated minimal lymphocytic infiltration (**D**,**I**,**N**,**S**). Statistical significance: *p* < 0.01 (**) and *p* = 0.0001 (****). Note: This panel focuses specifically on T-cell and B-cell markers.

**Figure 3 ijms-27-00197-f003:**
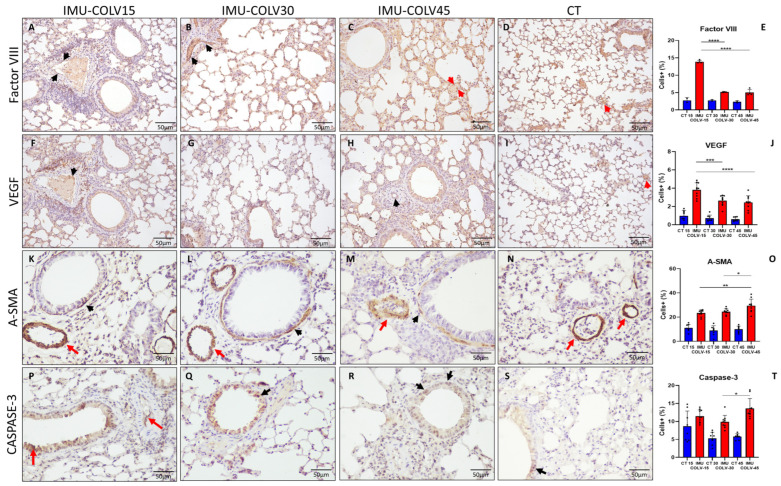
Vascular Reactivity in Lung Tissue of IMU-COLV Mice. Immunohistochemical analysis of vascular activation and remodeling in C57BL/6 mouse lungs following collagen V immunization. Factor VIII staining shows increased endothelial activation at 15 days (**A**), with expression declining at 30 and 45 days (**B**,**C**) to levels similar to controls (**D**), as quantified in (**E**). VEGF expression is highest at 15 days (**F**), localized primarily to the perivascular and peribronchial microvasculature, and progressively decreases at later timepoints (**G**,**H**), in relation to control (**I**), and with corresponding quantification in (**J**). *α-SMA* immunoreactivity is observed in smooth muscle layers of vessels and bronchioles across all groups (**K**–**N**), with increased thickness and intensity at 45 days (**M**), as shown in the quantitative analysis (**O**). Cleaved caspase-3 staining demonstrates a late increase in endothelial apoptosis (**P**–**S**), most prominent at 45 days, with quantification shown in (**T**). Red arrows denote vascular structures; black arrows indicate bronchiolar regions. Statistical significance: * *p* < 0.05; ** *p* < 0.01; *** *p* < 0.001; **** *p* = 0.0001.

**Figure 4 ijms-27-00197-f004:**
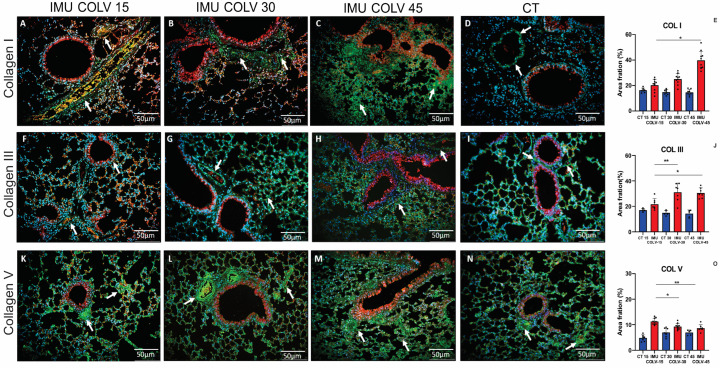
Remodeling of the Pulmonary Interstitium in the IMU-COLV Model. Immunofluorescence staining of collagen types I (**A**–**C**), III (**F**–**H**), and V (**K**–**M**) in lung tissue from C57BL/6 mice 15, 30, and 45 days after immunization with collagen V (IMU-COLV), compared with non-immunized controls (CT) (**D**,**I**,**N**). Collagens I, III, and V are shown in green (white arrows); nuclei in blue (DAPI); red blood cells in yellow; and tissue counterstaining in red–orange. Quantitative analysis shows progressive collagen accumulation in IMU-COLV groups (**E**,**J**,**O**) (* *p* = 0.0181; ** *p* = 0.0031). Specificity of labeling was verified using isotype IgG controls and no-primary-antibody controls, confirming that observed signals reflect true collagen immunostaining rather than nonspecific background or autofluorescence.

**Figure 5 ijms-27-00197-f005:**
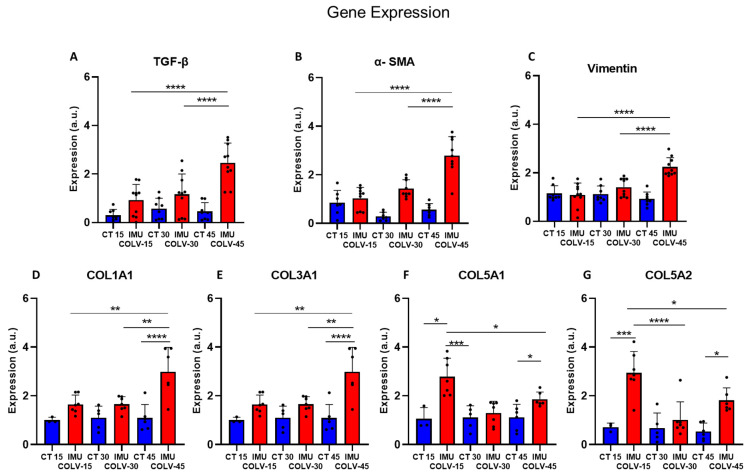
Relative gene expression of profibrotic markers in lung tissue. Gene expression levels of the *Tgfb1* (TGF-β) (**A**), *Acta2* (*α-SMA*) (**B**), *Vim* (Vimentin) (**C**), *Col1a1* (**D**), *Col3a1* (**E**), *Col5a1* (**F**) and *Col5a2* (**G**) were quantified by qPCR and normalized to the reference gene β2-microglobulin (*B2m*). Values are presented as relative expression, expressed in arbitrary units (a.u.), compared with the control group. Significant differences between experimental groups are indicated (* *p* < 0.05; ** *p* < 0.01; *** *p* < 0.001; **** *p* < 0.0001).

**Figure 6 ijms-27-00197-f006:**
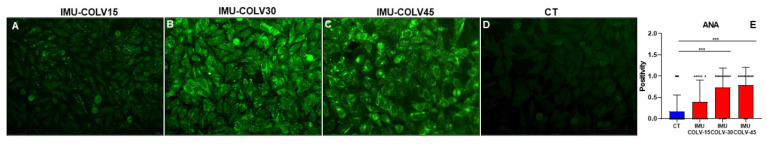
ANA Presence during Fibrosis Progression in the IMU-COLV Model. ANA research by immunofluorescence with Hep-2 cells in the progression of the IMU-COLV model, where it was predominantly negative in the serum of the control group (**D**) and at 15 days (**A**), while ANA research was positive in the 30 (**B**) and 45 (**C**) days groups. ANA showed a dotted pattern as the model progressed. Original magnification: 400×. GraphPad Prism version 8; One-way ANOVA, followed by Turkey and Sidak post-tests (*** *p* < 0.001) (**E**).

**Figure 7 ijms-27-00197-f007:**
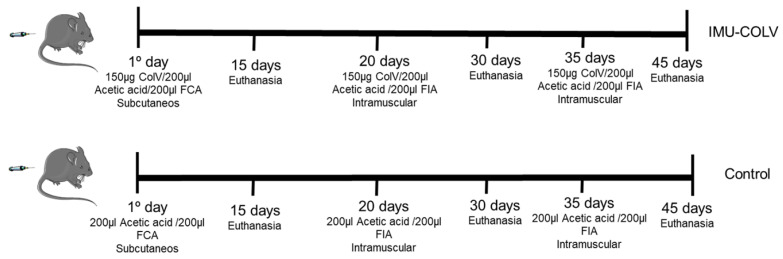
Schematic of the Col V immunization protocol for inducing the Ssc model in C57BL/6 mice and controls. Study groups were euthanized at 15 (*n* = 10), 30 *(n* = 10) and 45 (*n* = 10) days IMU-COLV: immunized with Col V. FCA: Freund’s complete adjuvant; FIA: Freund’s incomplete adjuvant; SC: subcutaneous; and IM: intramuscular.

**Table 1 ijms-27-00197-t001:** Sequence of oligonucleotides.

Gene	Sense 3′–5′	Antisense 5′–3′
*Gapdh*	ACA CAT TGG GGG TAG GAA CA	ACC CAG AAG ACT GTG GAT GG
*B2m*	CAT GGC TCG CTC GGT GAC C	AAT GTG AGG CGG GTG GAA CTG
*Col1a1*	GAG CGG AGA GTA CTG GAT CG	GCT TCT TTT CCT TGG GGT TC
*Col3a1*	GCA CAG CAG TCC AAC GTA GA	TCT CCA AAT GGG ATC TCT GG
*Col5a1*	GGT CCC TGA CAC ACC TCA GT	TGC TCC TCA GGA ACC TCT GT
*Col5a2*	CCT CAG GGA ATT GAT GGA GA	AGA GCC AGG CAT GAG TCC TA
*Acta2*	GTC GAT GTG CAG TGT GTG AG	CTC GCC CAT GAC ATT CGA TG
*Tgfb1*	ACT GCT TCC CGA ATG TCT GA	TCG CTT TGT ACA ACA GCA CC
*Vim*	CCT CCT GCA ATT TCT CTC GC	CGC TTT GCC AAC TAC ATC GA

## Data Availability

The datasets used and/or analysed during the current study are available from the corresponding author on reasonable request.

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
