# Peer review of "Temporal Dynamics of Pulmonary Fibrosis and Immune Dysregulation in a Collagen V-Driven Systemic Sclerosis Model"

_ijms, 2025, doi:10.3390/ijms27010197_

Round 1
Reviewer 1 Report
Comments and Suggestions for Authors
In this manuscript, Contini and colleagues investigated the COLV-immunized pulmonary fibrosis animal model in particular time course of its pathogenesis. The group has already published COLV immunized mouse study as a SSc animal model ( Arthritis Res Ther. 2019 Dec 11;21:278. doi: 10.1186/s13075-019-2052-2. Proposition of a novel animal model of systemic sclerosis induced by type V collagen in C57BL/6 mice that reproduces fibrosis, vasculopathy and autoimmunity) and reduce its novelty. However, this manuscript contributes new information to the field of SSc animal model research.
Comments.
- It was known that macrophages are a major cell type for driving lung fibrosis in animal models. Did the authors examine the macrophage population and types in the lung tissues? Figure 2 showed T lymphocytes and B cells, but not monocytes or macrophages.
- Figure 4: It is not clear what each color of green, red, blue, and yellow represents. Add this information to the Figure legend.
- Figure 5: Add information of gene expression (u.a.) meaning. Fold induction or absolute value?
Reviewer 2 Report
Comments and Suggestions for Authors
This article is devoted to the study of the collagen V-Driven SS model. The article's primary focus is on the early stages of damage, which should be emphasized in the title. This research may help better understand the pathogenesis of the disease and identify potential new targets for therapy. However, despite the article's relevance, a number of questions and concerns remain that need to be addressed before publication:
Major points
- In what tissues is collagen V normally found? Is it localized to specific tissue structures, e.g. the walls of large bronchioles, alveoli, blood vessels, etc.? Understanding collagen V localization will help better understand the mechanisms of fibrosis development and possible cell types involved in SS pathogenesis.
- Why were only mice females selected for the study?
- Interstitial lung diseases (ILD) are primarily characterized by the development of fibrosis in the alveoli. However, the representative images (Fig 1) demonstrate changes primarily in peribronchovascular region. Does fibrosis develop concurrently in the interstitial regins, or is interstitial fibrosis only observed in later stages? Furthermore, the representative images contain bronchioles of different sizes. Collagen deposition in the bronchiolar wall directly depends on its size. Images of bronchioles of comparable size are required.
- Why was human placental collagen COLV (Col V; Sigma) chosen for administration? Could the observed reaction be caused by a nonspecific immune response to a foreign human antigen?
- Fig 3 - the red arrows point to positive cells in both the bronchioles and blood vessels. This is confusing in the chapter "Endothelial Activation and Vascular Remodeling in the IMU-COLV Model." It is strongly recommended to expand the chapter and change the color of the arrows, separating the bronchiolar and vascular components.
- Figure 4 - the image contains at least three colors (red, blue, and green). All three colors must be describing in figure legend. Furthermore, the images have not background correction, preventing the specific staining from being visible, and the entire tissue appears homogeneously green. Background correction must be performed using relevant IgG controls, as well as auto fluorescence controls.
- A table with primers is missing.
- Information and representative photographs of the IgG used for immunohistochemistry must be added.
- How were the number of positive cells and nuclei counted? Several images show that the cells are close to each other, making accurate counting difficult. Were positive cells counted only in certain areas, such as VEGF-positive cells only in vessels? It is clear from the photographs that positive cells are also present in other areas. Counting the total number of positive cells across the entire section may significantly distort the results.
- In addition to myofibroblasts, activated stromal cells are also considered drivers of progressive fibrosis, including in the early stages. Is there evidence that this cell type is involved in the development of fibrosis in this model or in systemic sclerosis?
Minor points
- Figure 7 - I recommend describing the number of animals in each groups
- Line 537 - anti-Col V 537 (1:1,000) - absence of manufacturer
- It is known that fibrosis can develop unevenly in different lung lobes. Which lung part was collected for molecular analysis? Was this procedure standardized?
- The low image resolution does not allow resolve reliable image detail.
Round 2
Reviewer 2 Report
Comments and Suggestions for Authors
All comments have been corrected.